# Combinatorial design of molecular seeds for chirality-controlled synthesis of single-walled carbon nanotubes

Joerg Tomada [1], Thomas Dienel [2], Frank Hampel[1], Roman Fasel [2] & Konstantin Amsharov [1]

The chirality-controlled synthesis of single-walled carbon nanotubes (SWCNTs) is a major challenge facing current nanomaterials science. The surface-assisted bottom-up fabrication from unimolecular CNT seeds (precursors), which unambiguously predefine the chirality of the tube during the growth, appears to be the most promising approach. This strategy opens a venue towards controlled synthesis of CNTs of virtually any possible chirality by applying properly designed precursor molecules. However, synthetic access to the required precursor molecules remains practically unexplored because of their complex structure. Here, we report a general strategy for the synthesis of molecular seeds for the controlled growth of SWCNTs possessing virtually any desired chirality by combinatorial multi-segmental assembly. The suggested combinatorial approach allows facile assembly of complex CNT precursors (with up to 100 carbon atoms immobilized at strictly predefined positions) just in one single step from complementary segments. The feasibility of the approach is demonstrated on the synthesis of the precursor molecules for 21 different SWCNT chiralities utilizing just three relatively simple building blocks.

[1] Friedrich-Alexander-University Erlangen-Nuremberg, Department of Chemistry and Pharmacy, 91058 Erlangen, Germany. [2] Empa, Swiss Federal Laboratories for Materials Science and Technology, 8600 Dübendorf, Switzerland. Correspondence and requests for materials should be addressed to K.A. (email: konstantin.amsharov@fau.de)

Single-walled carbon nanotubes (SWCNTs) have been widely touted for their superior electronic properties and high potential to fulfil dreams in the realm of nanotechnology[1–12]. The SWCNT structure can be imagined as a graphene layer rolled up to a hollow seamless cylinder. The way how the graphene sheet is wrapped is defined by the pair of the so called chiral indices $n$ and $m$, which uniquely define the atomic structure of the nanotube[1]. The diversity in electronic properties of SWCNTs strongly depends on their chirality and can be divided into three types depending on the band structure. These include: truly metallic armchair-SWCNTs with zero band gap ($n-m = 0$), quasi-metallic ($n-m = 3q$, where $q$ is a nonzero integer) with a very small band gap, and semiconducting ($n-m \neq 3q$, where $q$ is an integer) tubes with band gaps up to 1.5 eV[1–3]. Due to outstanding mechanical properties, chemical robustness and the possibility to tune the band gap over a wide range, SWCNTs are considered to be the most promising candidates for the potential future nanoelectronics[4–12]. However, the widespread application and exploration of the potential of these fascinating carbon materials are impeded by the limited availability of uniform chirality-pure samples. SWCNTs are currently produced by poorly controlled methods, which provide a highly heterogeneous mixtures of various chiralities and diameters[13,14]. The successful tackling of this problem is expected to revolutionize the science of carbon-based nanomaterials[15]. Among various sophisticated approaches, suggested for the chirality-controlled synthesis of SWCNT[15–17], the use of molecular templates which unambiguously dictate the chirality of the SWCNTs during the growth, holds the most potential[17–21]. However, until recently all attempts to put this long standing challenge into practice were rather unsuccessful, showing insufficient selectivity and poor control over the SWCNT structure[17–34]. A breakthrough in this field was achieved by Amsharov and Fasel et al., who demonstrated the chirality-controlled synthesis of SWCNTs by epitaxial elongation of well-defined ultra-short nanotubes used as molecular seeds[35]. This method currently remains the only one, which allows truly rational control over SWCNT chirality and provides unprecedentedly high-chiral purity[15,17]. The respective atomically precise short SWCNTs (CNT caps) were fabricated by bottom-up strategy applying surface-assisted zipping of specially preprogrammed precursors[36–40]. Among the possibility to perform several challenging steps (CNT cap synthesis, seed deposition, seed activation and CNT growth) as a one-pot process (Fig. 1), the approach opens a path towards rational synthesis of SWCNTs of desired chirality as soon as required precursor molecules become available. To reach the full potential of this method, effective synthetic routes to SWCNT precursors must be developed.

Here we demonstrate a general strategy for the synthesis of precursors for controlled growth of SWCNTs with various predefined chiralities. The suggested combinatorial design allows facile assembly of SWCNT precursors from simple complementary cyclic ketones, including precursors for highly interesting semiconducting tubes.

## Results

**Combinatorial approach.** Accepting that only IPR (Isolated Pentagon Rule, stating that all pentagons have to be completely surrounded by hexagons in order to form less strained system[41],) structures are stable, all possible SWCNT-caps (for a certain chirality) can be derived by constructive enumeration[42]. Further, the suitable precursor structures can be found retrosynthetically from the respective SWCNT end-caps by formal unrolling to the quasi-planar PAH (polycyclic aromatic hydrocarbon)[43,44]. The last one can be obtained by multi-step organic synthesis. However, this way appears to be very impractical, since each particular SWCNT-chirality would require developing of individual synthetic routes to

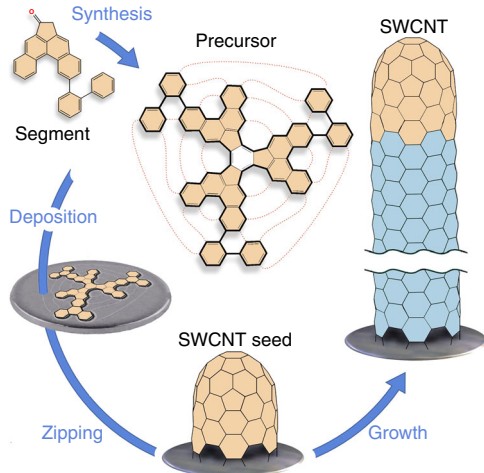

**Fig. 1** Schematic representation of the first rational surface-assisted synthesis of (6,6) SWCNTs[36]. The main steps are illustrated: precursor synthesis, precursor deposition on the metal surface, catalytic cyclodehydrogenation (zipping) to the SWCNT seed and the subsequent epitaxial elongation of the target chirality pure SWCNT

the respective PAH-precursor. Note, the synthesis of such extra-large PAHs is far from trivial, and frequently associated with serious synthetic problems mainly connected with solubility issues, which especially relates to the low symmetry systems whose total synthesis appears to be extremely challenging. Thus, the aldol cyclotrimerization of cyclic PAH-ketones (Fig. 1) proven to be a prolific approach to extra-large PAH-precursors[19,25,29,35,45–52], is limited to $C_3$ symmetrical systems, which could cover only a tiny fraction of possible chiralities ($n = 3a$, m = $3b$; where $a$ and $b$ are integers). Since all $C_3$ symmetrical precursors gave exclusively metallic nanotubes the synthesis of precursors for highly interesting semiconducting tubes remains elusive. Here we present a general synthetic concept of the precursor engineering for SWCNT-caps by combinatorial multi-segmental assembly, that provides simple access to SWCNT precursors for virtually any desired connectivity.

The concept of combinatorial design is outlined in Fig. 2. Formally, any possible SWCNT end-cap can be represented as a truncated cone, which surface can be cut in several segments as shown in Fig. 2a. The top base area (highlighted in blue) serves as a branching unit which connects all segments (highlighted in orange). The cutting procedure of the cap can be imagined by the formal cleavage of C–C bonds interconnecting the segments. This procedure is similar to the retrosynthetic analysis described above except for that all segments should be connected through the central unit. Regardless of the cutting, the periphery of each produced segment can be described by the individual segmental chiral vector $C_h{}^S$ which contributes to the final chiral vector $C_h$. Thus, the sum of chiral vectors of all segments will result in vector $C_h$ (Fig. 2b). Therefore, each segment can be assigned using two chiral indexes $n_S$ and $m_S$, similarly to the indexation of SWCNTs. Consequently, the final chirality ($n$, $m$) of the SWCNT-cap is simply determined by the sum of $n_s$ and $m_s$ indexes of all incorporated segments (Fig. 2c). Since there are many alternative ways for the cap segmentation, many different segments with various chiralities can be generated. Several very simple segments are shown in Fig. 2d as an example. The power of the multi-segmental assembly strategy lies in the possibility to combine different segments (independently on the sequence) which will yield a hemispherical SWCNT-cap with additive chirality (Fig. 2e). Note, that from the mathematical viewpoint this scheme works without exceptions, and so far, we could not locate any limitations. However, not all combinations are practically meaningful, since the fusion of segments could lead to

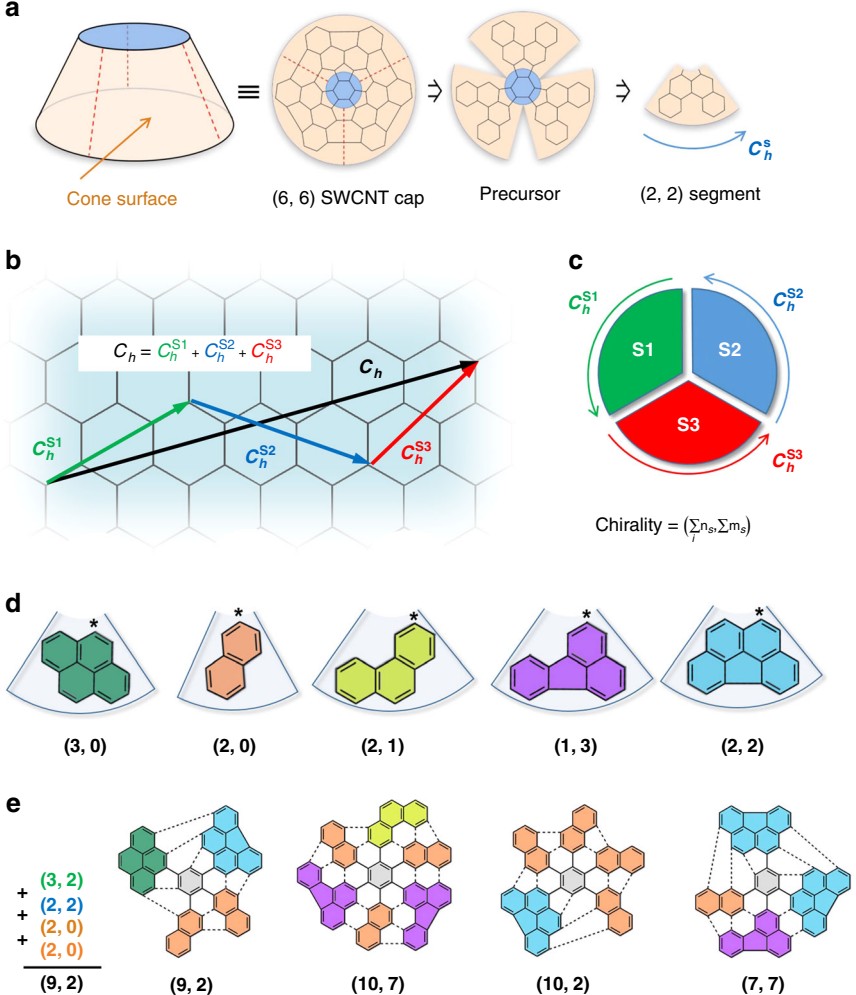

**Fig. 2** Concept of end-cap engineering by multi-segmental assembling. **a** Cap segmentation on the example of division of a (6,6) CNT cap into three equal (2,2) segments; **b** Determination of the chirality of SWCNT cap with the help of vectors; **c** schematic representation of contribution of segmental chirality vectors to the final vector $C_h$; **d** several possible complimentary segments displayed with their segmental chirality, the asterisk indicates the attachment position to the central branching unit; **e** examples of multi segmental assembly of CNT caps with various chiralities. Note, that PAH structures shown are not suitable as SWCNT precursors because of the conformational mobility of segments. Required rigidity can be achieved by additional inter-segmental connections which are not shown here for clarity

non-realistic geometries from a chemist's point of view, such as pentagon–pentagon junctions or four-membered rings. Nevertheless, it is possible to select the so called complementary segments whose fusion will produce stable IPR-geometries. Thus, having only few segments a big number of IPR-SWCNT-caps with various chiralities can be constructed as it is schematically shown in Fig. 2e (only few combinations are given as an example).

**Precursor synthesis**. To demonstrate the feasibility of the segment-assembly strategy we have chosen three complementary segments **A** (2,2 segment), **B** (3,1 segment) and **C** (3,0 segment), which can be assembled by means of aldol condensation to the respective trimers[35,45–49], or tetramers[51]. The synthetic routes to **A** (**A′**), **B** and **C** are summarized in Fig. 3. In all cases the key cyclopentanone fragment was introduced by the radical bromination of the benzylic position of the respective methylarene followed by cyanation and hydrolysis to the corresponding carboxylic acid which was converted to the ketone by intramolecular Friedel-Crafts acylation. In the case of **A′** the respective methylarene was obtained by Suzuki-Miyaura coupling of biphenyl-2-boronic acid and 1-bromo-4-methylnaphthalene followed by Mallory-photocyclization. Segment **A** was prepared directly by

standard Suzuki-Miyaura coupling of **BrAN** (5-bromoace-naphthenone) and biphenyl-2-boronic acid. Effective synthesis of the block **B** by coupling of **BrAN** and 9-anthraceneboronic was possible after protective reduction of carbonyl group to alcohol or alternatively by applying optimized Suzuki-Miyaura conditions. Precursor **C** was synthesized starting from methyltriphenylene[25]. After bromination of the benzylic position followed by Michaelis-Arbuzov reaction with triethylphosphite the respective arylmethyl phosphonate was used in the Wittig-Horner reaction with 2-methylbenzaldehyde yielding corresponding diarylethane. The following Mallory photocyclization resulted in the formation of the desired methylbenzohelicene which was obtained in pure form after several repetitive recrystallizations from xylene. Final introduction of the cyclopentanone fragment was performed according to the general procedure discussed above (for synthesis details see Supplementary Methods).

As it is shown in Fig. 3, the segment **A** was prepared in two modifications **A** and **A′**. The segment **A** with the flexible biphenyl moiety was synthesized with the aim to improve the solubility of final precursors, since the solubility is a necessary prerequisite for the following separation and characterisation. Similar to **A**, the anthracene moiety in segment **B** was not fused with the

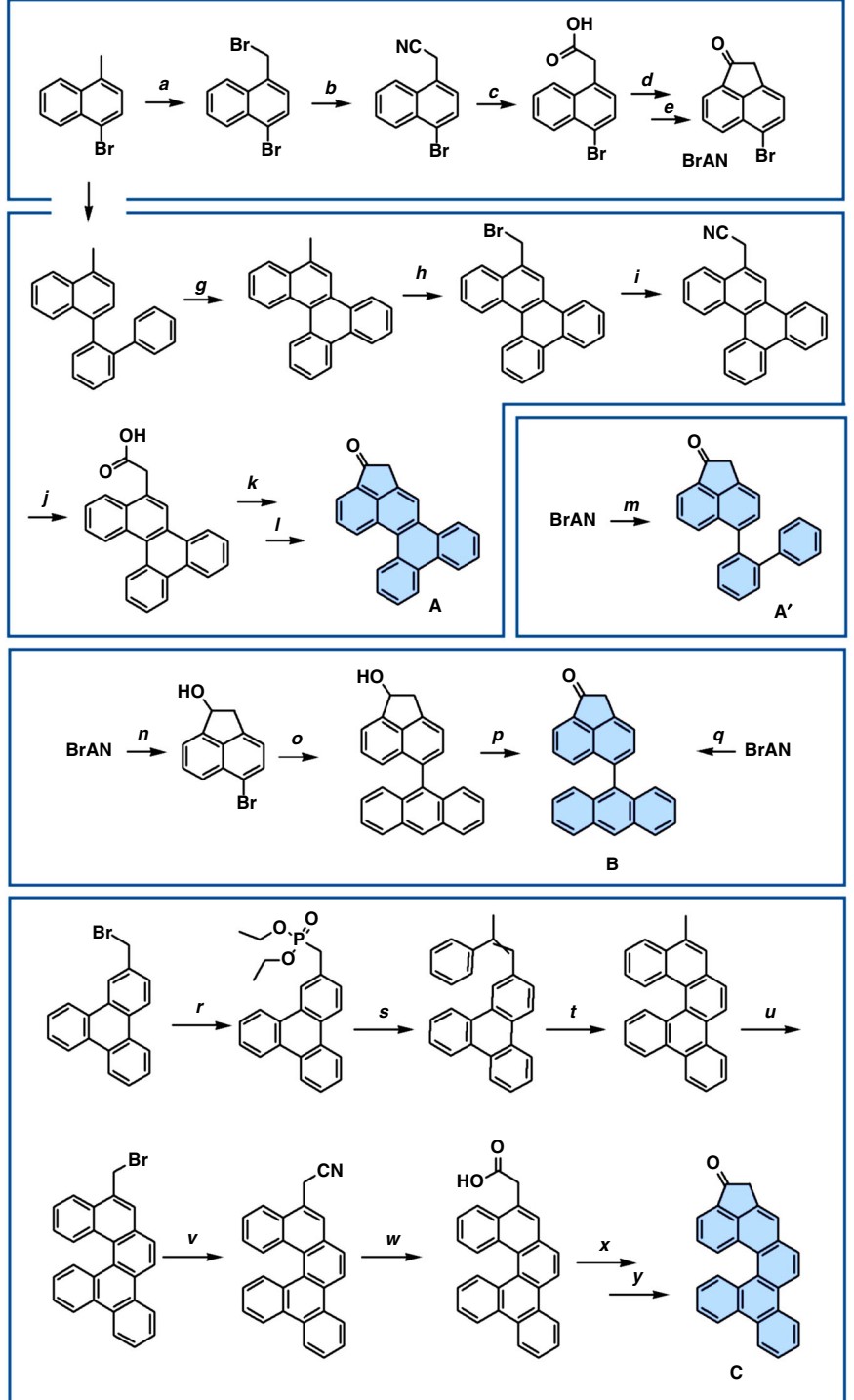

**Fig. 3** Synthesis of segments **A**, **A'**, **B** and **C**. *a*) NBS, DBPO, DCM, reflux; *b*) KCN, TBAB, DCM, H$_2$O, 12 h at RT, 89%; *c*) H$_2$SO$_4$, H$_2$O, HOAc, 12 h reflux, 96%; *d*) SOCl$_2$, 90 min reflux, 65 °C; *e*) AlCl$_3$, CH$_2$Cl$_2$, 1 h at 0 °C then 15 min reflux, 65%; *f*) 2-biphenyl boronic acid bis(pinacol) ester, Pd(PPh$_3$)$_4$, K$_2$CO$_3$, toluene/MeOH (3:1), 12 h reflux, 64%; *g*) I$_2$, *hv*, propylene oxide, cyclohexane, 5 h, 90%; *h*) NBS, DBPO, CCl$_4$, 2 h reflux, 68%; *i*) KCN, TBAB, DCM, H$_2$O, 3 h at RT, 94%; *j*) H$_2$SO$_4$, H$_2$O, HOAc, 12 h reflux, 98%; *k*) C$_2$O$_2$Cl$_2$, 1 h reflux; *l*) AlCl$_3$, CH$_2$Cl$_2$, 1 h at 0 °C then 15 min reflux, 78%; *m*) 2-biphenyl boronic acid bis(pinacol) ester, Pd(PPh$_3$)$_4$, K$_2$CO$_3$, toluene/MeOH (3:1), 12 h reflux, 72%; *n*) NaBH$_4$, MeOH/THF (1:1), 2 h at RT, 98%; *o*) 9-anthraceneboronic acid bis(pinacol) ester, Pd(PPh$_3$)$_4$, Cs$_2$CO$_3$, toluene/MeOH (3:1), 12 h reflux, 68%; *p*) PCC, DCM, 1 h at RT, 46%; *q*) 9-anthracene boronic acid, K$_2$CO$_3$, Pd (PPh$_3$)$_4$, THF/H$_2$O (2:1), 24 h reflux, 64%; *r*) P(OEt)$_3$, 160 °C; *s*) acetophenone, KOtBu, THF, 50 °C; *t*) I$_2$, *hv*, propylene oxide, cyclohexane, RT; *u*) NBS, DBPO, DCM, reflux, 86%; *v*) KCN, TBAB, DCM, H$_2$O, RT, 86%; *w*) H$_2$SO$_4$, H$_2$O, HOAc, reflux, 94%; *x*) C$_2$O$_2$Cl$_2$, 1 h; *y*) AlCl$_3$, CH$_2$Cl$_2$, 1 h at 0 °C then 15 min reflux, 68%

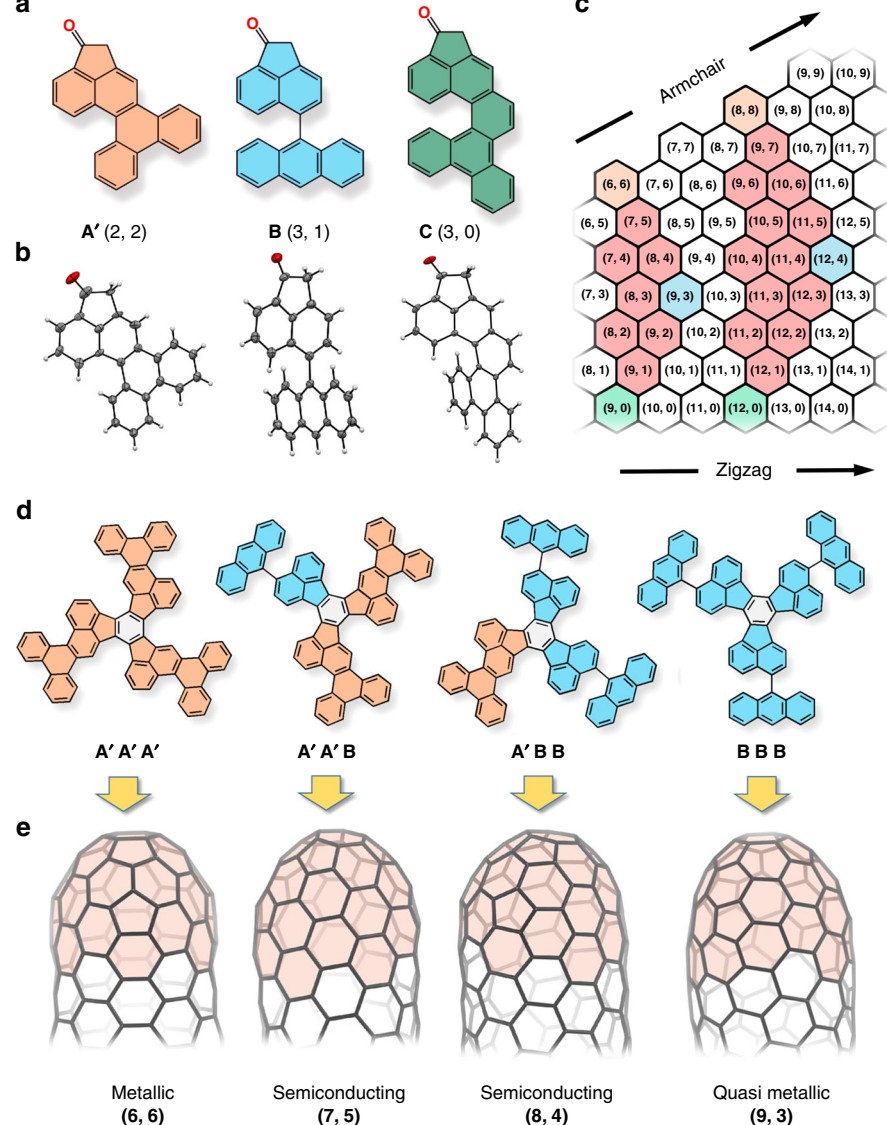

**Fig. 4** Segmental assembly of cyclic ketones **A'**, **B** and **C**. **a** The complementary segments with different segmental chirality; **b** ORTEP plot of segments **A'**, **B** and **C**. Thermal ellipsoids are set at 50% probability level; **c** chirality map displaying the chiralities possible to obtain out of the displayed segments; **d**, **e** possible trimers formed by assembling of segments **A'** and **B** and corresponding CNT-caps (highlighted in red) and SWCNTs

acenaphthenone part in order to provide enhanced solubility of the final products. Note that conformational mobilities of the biphenyl and anthracene moiety do not affect the final connectivity of the CNT cap. The relatively enhanced solubility of the segment **C** was achieved by the steric congestion in the helicene-like motif leading to perturbation of the inherently planar π-system. The geometries of all building blocks (Fig. 4a) were visualized by means of single crystal X-ray analysis (Fig. 4b). All segments were specifically designed to have a different number of carbon atoms ($C_{24}H_{14}$, $C_{26}H_{14}$ and $C_{28}H_{14}$ for **A**, **B** and **C** accordingly) allowing simple identification/characterisation of final precursors by MS analysis.

The combinatorial assembly of these three simple ketones could provide SWCNT precursors for 25 different chiralities (10 for the trimer and 15 for the tetramer), which are outlined in Fig. 4c. Thus, the combination of **A** and **B**, will result in the statistical formation of four trimers, namely **AAA**, **AAB**, **ABB** and **BBB**, which are precursors for (6,6), (7,5), (8,4), and (9,3) SWCNTs (Fig. 4d). The expected products of tandem cyclodehydrogenation (SWCNT-caps) and respective tubes are shown in Fig. 4e. Importantly, the

regiospecificity of each intramolecular cyclization step is pronouncedly predefined by the rigid structure of the precursor molecule. The on-surface cyclodehydrogenation of such rigid PAH systems has been proven to occur with close to quantitative yield[35–40]. All segments are designed in such a way, that the final SWCNT-caps (highlighted in red in Fig. 4e) already contain several completed SWCNT-belts which are exclusively composed of hexagons (with no pentagons on the periphery). Such ultra-short SWCNTs possess enhanced stability and are expected to be very effective for the initiation of the nanotube growth under CVD condition[29,35,44].

The last synthetic step, the conversion of ketones to cyclic trimers/tetramers, was performed via aldol cyclomerization in *o*-dichlorobenzene (ODCB) using either Lewis or Brønsted acid. The outcome of the reaction using *p*-toluenesulfonic acid at 160 °C[22], was rather disappointing since the reaction was almost fully halted after dimer formation. Similar results were previously observed for the rigid and badly soluble ketones, this observation was connected to the low solubility of the dimer[25]. Applying the protocol optimized for the hardly soluble trimers[51], using ODCB/TiCl₄ system and heating of the reaction mixture in a sealed glass ampoule at 180 °C

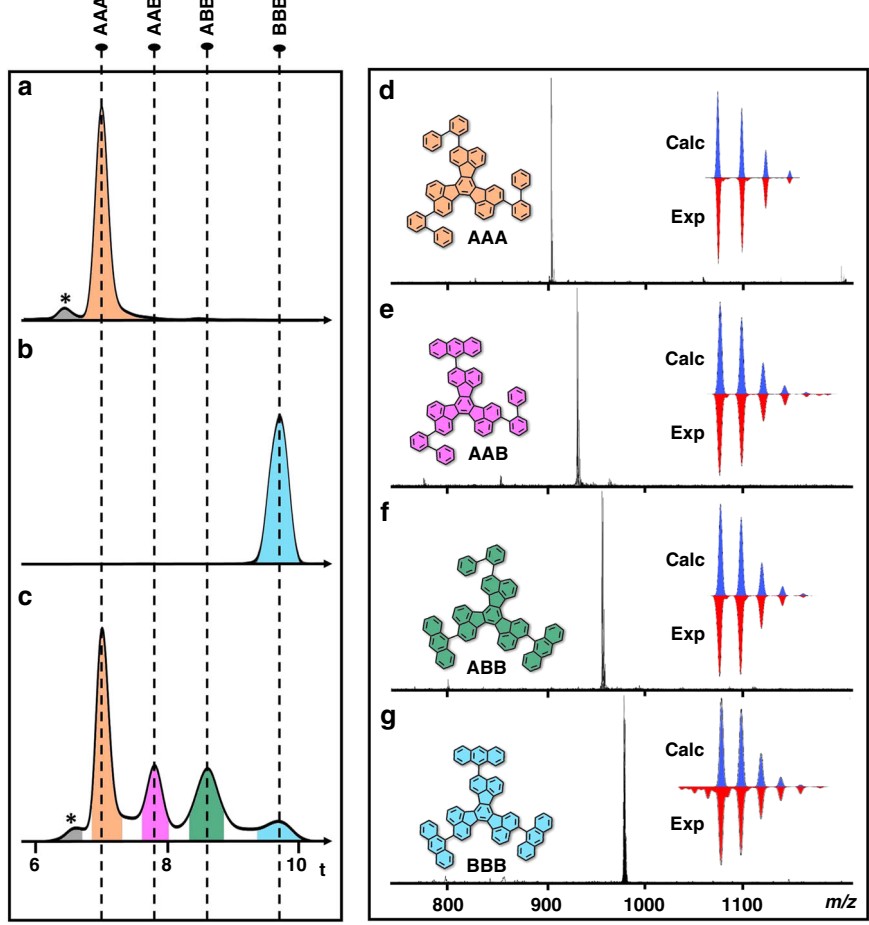

**Fig. 5** HPLC/MS analysis of the cyclotrimerization products. **a** HPLC profile of trimer **AAA** as obtained. The corresponding tetramer **AAAA** is marked with asterisk; **b** HPLC profile of trimer **BBB** as obtained; **c** HPLC profile of reaction mixture of combinatorial assembly of segment **A** and **B** in a 1:1 ratio showing formation of four trimers. The corresponding tetramers are marked with asterisk; **d–g** MS analysis of the separated trimers. Experimental and theoretical isotopic distribution are shown in inset. HPLC separations are performed on COSMOSIL-5PBB column 10 × 250 column (eluent toluene/MeOH 3:2, flow rate 5 mL min$^{-1}$, $t$ = 40 °C, detection at 380 nm)

for 24 h, gave the desired trimer **AAA** as a major product according to HPLC analysis. However, the cyclization of **B** under these conditions was accompanied by the partial decomposition, leading mainly to the loss of the anthracene moiety, as indicated by the HPLC/MS analysis. Therefore, further optimization of the reaction conditions was performed for the more labile block **B**. The condensation process was investigated in the ODCB/TiCl$_4$ mixture at the temperature region of 80–160 °C (20 °C step) for 0.5–10 h. It was found that the reaction can be accomplished under relatively mild condition similar to that reported by Scott et al. for the synthesis of a C$_{60}$ fullerene precursor[45]. The best conversion of **B** to trimer **BBB** was observed at 120 °C after 2 h. Under optimized conditions both **A** and **A′** were successfully converted to the respective homo-trimer. Next, ketone **A** was trimerized under optimized conditions on a 50 mg scale. All polar by-products including dimers, acyclic oligomers and starting ketone were removed by flash chromatography on silica using toluene as an eluent yielding practically pure **AAA** in close to 70% yield. The HPLC analysis of the sample reveals rather clean transformation to **AAA** and only trace amounts of the tetramer **AAAA** (Fig. 5a). Cyclotrimerization of **A′** (10 mg scale) gave a badly soluble trimer in 60% yield and it was characterized by HPLC/MS. Similarly, ketone **B** was successfully converted to the trimer **BBB** in deca-milligram scale, and its solubility was found to be sufficient for further analysis. The isolated yield of **BBB** was found to be remarkably lower (30%),

this is probably connected with the partial oligomerisation of **B**. Interestingly, no tetramer **BBBB** was detected in this case as indicated by the HPLC analysis (Fig. 5b). Since the condensation conditions were found to be efficient for both blocks, we performed the combinatorial cyclotrimerization of **A** and **B** in equimolar ratio on a 50 mg scale. After removal of polar by-products the reaction mixture was analysed by MS showing the formation of all desired trimers. Due to sufficient solubility of all trimers the preparative HPLC separation appears to be possible (Fig. 5c). It was found that the elution times correlate with the molecular weight and each individual trimer can be isolated in pure form by the single HPLC step (Fig. 5). All four trimers were obtained in multi mg scale allowing their characterization by $^1$H-NMR, UV–Vis spectroscopy (see "Supplementary Methods") and MS spectrometry (Fig. 5d–g). The combination of **B** and **A′** gave essentially the same results however the separation and analysis of products was hindered because of remarkably lower solubility of products. Therefore, all further assemblies were performed using block **A** exclusively. Further we have performed a combinatorial assembly of blocks **A** and **C** and found it to be successful according to MS analysis. However, the helical structure of **C** did not provide sufficient contribution to the final solubility and only trimer **AAC** was found to be soluble enough for HPLC separation whereas **ACC** and **CCC** were obtained as virtually insoluble compounds. Similarly, the combination of **B** and **C** gave acceptable soluble **BBC** while **BCC**

and **CCC** were obtained as hard-to-separate mixtures (see "Supplementary Methods"). Moreover, partial cyclodehydrogenation in the fjord region of the block **C** was observed in both cases. Although this reaction does not affect the chirality of the precursor the separation of individual trimers becomes much more complicated. Thus, among ten possible trimers which can be theoretically assembled from **A**, **B** and **C** blocks, we have synthesized nine (all except **ABC** combination) and were able to separate six of them in pure form. Four of which (**AAA**, **AAB**, **ABB** and **BBB**) were obtained in multi milligram amounts, with an overall yield of 49% for all four trimers (**AAA** = 41%, **AAB** = 20%, **ABB** = 29% and **BBB** = 10% at a combination ratio of segment A to B of 1:1). Note, that after cyclodehydrogenation on a catalyst surface, the respective CNT seeds can be elongated a multi-thousand fold during the epitaxial growth, thus necessitating only milligram amounts of the precursor for the multigram or even kilogram scale production of isomer pure SWCTs.

Finally, we have performed aldol cyclotetramerization of ketones to demonstrate the general feasibility of our strategy.

The cyclotetramerization was carried out in a ODCB/TiCl$_4$ mixture specially optimized for the tetramer formation[53]. However, the reaction cannot be shifted completely to the tetramer formation and it is always accompanied by the formation of trimers. The combinatorial tetramer synthesis was examined on the segmental assembly using blocks **A** and **B** and resulted in the formation of a rather complex mixture containing six tetramers and four trimers as indicated by the MS analysis. Noteworthy, that only five signals for the tetramer can be observed in the MS spectrum since isomeric tetramers **AABB** and **ABAB** cannot be distinguished by the MS analysis. However, both tetramers are precursors for the same chirality (10,6) since the final chirality is determined as the sum of segmental chirality and thus it is not affected by the sequence of segments. Further in the text the abbreviation **A2B2** indicates the composition and will be used for both isomers. Despite the complex composition of the reaction mixture it was possible to perform analytical separation and identification of all individual molecules applying gradient HPLC/off-line MS analysis (Fig. 6). Among trimers and tetramers

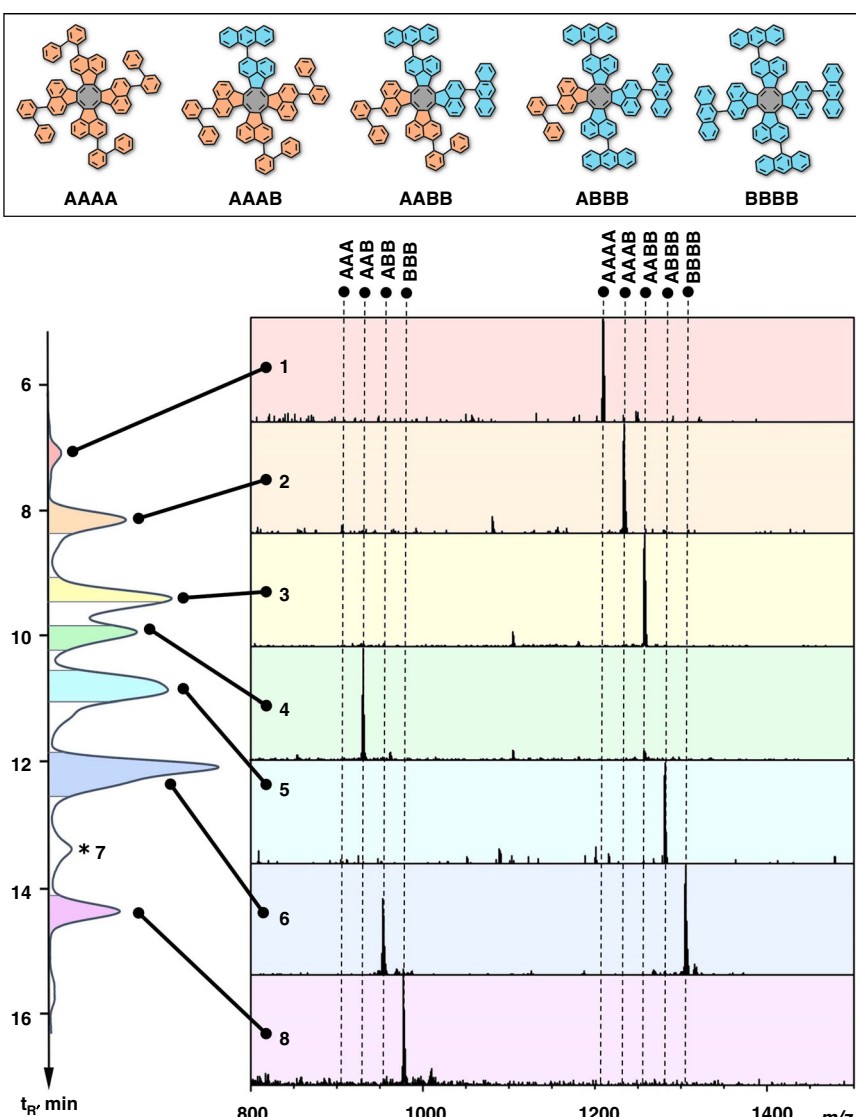

**Fig. 6** HPLC/MS analysis of the combinatorial cyclotetramerization of segments **A** and **B** (1:2 ratio). Fractions 1, 2, 4, 5 and 8 contain individual precursor molecules **AAAA**, **AAAB**, **AAB**, **ABBB** and **BBB** respectively. Fraction 3 contains both isomers for the (10,6) chirality (**AABB** and **ABAB**). The molecular structure is given for only one isomer **A2B2 (AABB)**. Fraction 6 contains the trimer **ABB** and the tetramer **BBBB**; Fraction 7 (marked with asterisk) contains the mixture of several hexamers. HPLC separation is performed on COSMOSIL-PBr column 4.6 × 250 (eluent DCM/MeOH 4:1, flow rate 1 mL min$^{-1}$, $t$ = 35 °C, detection at 380 nm)

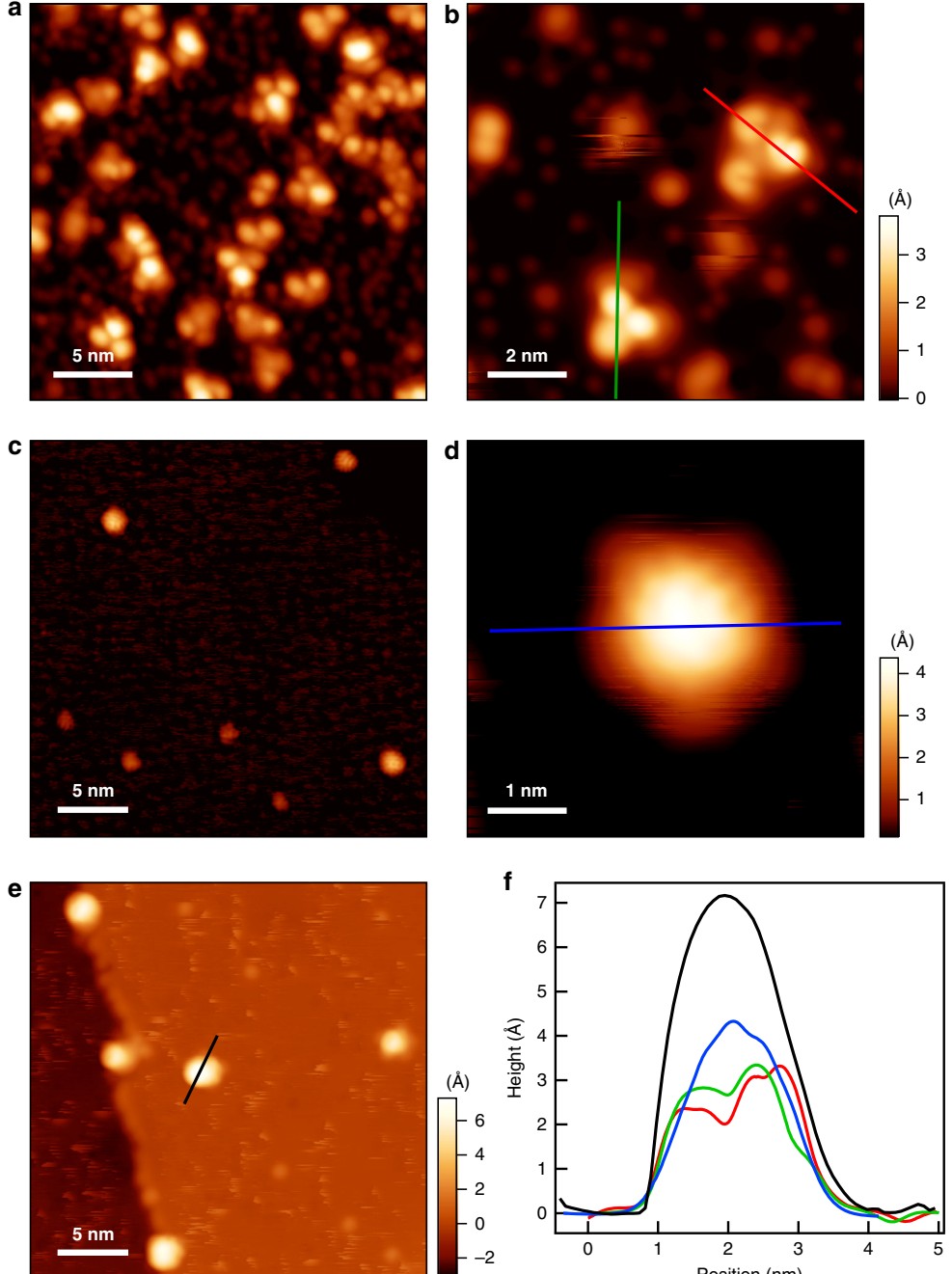

**Fig. 7** STM images of the formation of (7,5) SWCNT seeds from the **AAB** precursor. **a**, **b** Precursor molecules as deposited on Pt(111) held at room temperature. Molecules deposited on hot Pt(111) (**c**, **d**, held at 470 K), and after continuous annealing for 10 min at 470 K (**e**). **f** Line profiles (positions indicated in (**b**, **d**, **e**) over as-deposited precursors (red and green lines), closing cap (blue line), and a finished SWCNT seed (black line). STM setpoints are −0.4 V, 10pA (**a**), 1 V, 50pA (**b**), 1.5 V, 50pA (**c**), −3V, 100pA (**d**), and −2.5 V, 100pA (**e**)

the formation of small amounts of hexamers was detected whereas heptamers and pentamers are practically absent in the reaction mixture. Importantly, all mixed tetramers (**AAAB**, **A2B2** and **ABBB**) could be isolated in pure form (homo-tetramers **AAAA** and **BBBB** are easily accessible by the cyclotetramerization of individual blocks). In a similar way the cyclotetramerization was found to be effective for synthesis of tetramers via assembly of **A–C** and **B–C** blocks as indicated by the MS analysis. A table documenting all the SWCNT precursors synthesized in this work can be found in Supplementary Table 3.

**STM experiments**. To demonstrate the efficiency of precursors in the SWCNT nucleation the (7, 5) SWCNT precursor (**AAB**) was selected for scanning tunnelling microscopy (STM) investigation which is summarized in Fig. 7. As a first step the designated **AAB** was deposited by organic molecular beam epitaxy under ultrahigh vacuum conditions onto a clean Pt(111) surface. Analysis of a partially covered surface showed that the majority of the as-deposited molecules exhibits the quasi three-fold symmetric configuration. The molecules display shape and size expected for **AAB**, thus confirming that the precursor remains intact during deposition.

Note that one of the three distinct lobes of each molecule displays larger apparent height which is in line with the **AAB** structure (Fig. 7a, b). Deposition on a hot Pt surface (470 K) reveals that all quasi-planar **AAB** precursors immediately transform into dome-shaped species with a prominent increase in apparent height from 3 to 4 Å (Fig. 7c, d). The fully formed cap shows a remarkable further increase of apparent height up to 7.0 Å without change in diameter. (Fig. 7e, f). These results clearly demonstrate the successful formation of the targeted (7,5) SWCNT seed.

In summary we present a facile synthetic strategy to precursor molecules for the surface-assisted fabrication of isomerically pure SWCNTs, applicable for the synthesis of virtually any desired chirality. The synthetic path is based on the preparation of relatively simple complimentary segments which can be assembled to the required precursor. The general feasibility of the approach is demonstrated on the example of aldol cyclomerization of cyclic ketones. As a proof-of-concept, we demonstrate the synthesis of precursors for 21 different SWCNT chiralities using just three simple segments. The possibility to derive the required precursors in pure form was shown for 10 examples (six of which are unsymmetrical). All precursor molecules presented here exhibit rigid connectivity ensuring their correct cyclization to the CNT seed on a catalytic surface. The activity of precursors in a controlled nucleation is demonstrated at the selected example of (7, 5) SWCNT cap formation. The power of our approach lies in the ultra-simple retrosynthetic design and facile one-step assembly of extra-complex and until now elusive SWCNT precursors. We truly believe that our synthetic methodology towards synthesis of well-defined nanotube seeds for controlled growth of chirality pure SWCNTs will substantially contribute to the blooming field of carbon-based nanomaterials in both chemical and physical sciences.

## Data availability

All data are available in the article and its supplementary materials. Supplementary methods, details of STM experiments and chemical compound information are available in the supplementary materials. Crystallographic data CCDC no: 1858611, 1858612 and 1858613 can be obtained free of charge from The Cambridge Crystallographic Data Centre via www.ccdc.cam.ac.uk/data_request/cif

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

## Acknowledgements

Funded by the Deutsche Forschungsgemeinschaft (DFG) – Projektnummer 182849149 – SFB 953, AM407.

## Author contributions

K.A. and J.T. conceived the concept of the project and carried out the experiments and analysis. F.H. performed X-ray crystal structure analysis. T.D. and R.F. performed the STM study, J.T. and K.A. prepared the manuscript.

## Additional information

**Competing interests:** The authors declare no competing interests.

