## [Peer Review File · Nature Communications]

Reviewers' comments:

Reviewer #1 (Remarks to the Author):

The work described here is not suitable for publication in Nature Communications at this stage. The authors propose synthesizing unsymmetrical decacyclene derivatives (benzene rings fused on every other side by substituted acenaphthylene units, e.g., figure 3d) by performing acid catalyzed aldol cyclomerizations on mixtures of two or three substituted acenaphthenones (e.g., figure 3a). They speculate that the large polycyclic aromatic hydrocarbons (PAH) thus obtained could serve as precursors for the synthesis of chirality-pure single walled carbon nanotubes (SWCNT, e.g., figure 3e), using known technology (figure 1).

To demonstrate the feasibility of this proposal, the authors investigate the acid catalyzed aldol cyclomerizations of mixtures of two or three of the substituted acenaphthenones shown in schema 1. Despite a claim in the manuscript that "The possibility to derive required precursor in pure form was shown on 12 examples," the supplementary information reports NMR characterization of only two unsymmetrical decacyclene derivatives. Two others are reported with only HRMS characterization, and no others are reported. The approach suffers from serious problems caused by product insolubility, which hampers (or precludes) isolation, purification, and characterization of the expected unsymmetrical decacyclene derivatives. Thus, the realization of this proposed concept is still in a very preliminary state and is hardly useable. No attempt to convert any of the synthesized "precursors" into the corresponding SWCNTs is reported.

Schema 1 outlines a considerable amount of synthetic organic chemistry that the authors performed to access the substituted acenaphthenones for use in the acid catalyzed aldol cyclomerization reactions. Although such work did indeed provide the desired compounds, the chemistry used is not at all novel. The authors do not hold up this part of the work as a main selling point.

Two points should be corrected when the authors revise this manuscript:

1. In the Results and discussion, line 2: the authors say "...in order to from strainless system." In reality, even IPR systems are not "strainless"
2. In discussing the combinatorial trimerization of A and B, the authors say "...showing the formation of all desired mixed A/B trimers" They should say both of the desired mixed A/B trimers, since only two of the trimers are mixed.

Reviewer #2 (Remarks to the Author):

Tomada et al reported the synthesis of SWCNT precursors by aldol-type trimerization. Although the synthetic route is not new, I think this work is an important paper that will have a large impact on the field of carbon-based materials.

As a background, they accomplished controlled synthesis of single chirality CNT (Nature 2014). They synthesized C₉₆H₅₄ precursor, which was treated with surface-catalyzed cyclodehydrogenation. Then, epitaxial elongation of ultrashort nanotubes provided single chirality CNT. In order to extend this methodology, various precursors are naturally required.

The overall synthesis is classical that utilizes methyl group on the aromatic ring as a handle for further functionalization to make the corresponding ketones, ie. radical benzylic bromination/cyanation/hydration/Friedel-Crafts reaction. The ketones are treated under acidic conditions for trimerization. Under some conditions, the tetramers were obtained. Note that such

aldol-type trimerization that form benzene ring is well-known. Thus, this synthesis is essentially not new from the synthetic point of view. However, I think this work nicely established the route for various SWCNT precursors, which eventually opened up the possibility for programmed preparation of a series of single chirality SWCNT. Therefore, I recommend the manuscript to be published in Nature Comm. as a communication, which would attract not only organic chemist but also researchers who are working on carbon-based materials.

In the paper, there are many typos and mistakes. The authors should check carefully and correct them before submission.

Answer to Referees

Title: " Combinatorial Design of Molecular Seeds for Chirality Controlled Synthesis of SWCNTs "

Reviewer #1 (Remarks to the Author):

We appreciate the critical assessment made by Reviewer#1 which help us to improve the manuscript.

1. The authors propose synthesizing unsymmetrical decacyclene derivatives (benzene rings fused on every other side by substituted acenaphthylene units, e.g., figure 3d) by performing acid catalyzed aldol cyclomerizations on mixtures of two or three substituted acenaphthenones (e.g., figure 3a). They speculate that the large polycyclic aromatic hydrocarbons (PAH) thus obtained could serve as precursors for the synthesis of chirality-pure single walled carbon nanotubes (SWCNT, e.g., figure 3e), using known technology (figure 1). To demonstrate the feasibility of this proposal, the authors investigate the acid catalyzed aldol cyclomerizations of mixtures of two or three of the substituted acenaphthenones shown in schema 1.

We agreed on this point and have included the demonstration of the SWCNT growing from the one of synthesised precursor.

2. Despite a claim in the manuscript that "The possibility to derive required precursor in pure form was shown on 12 examples," the supplementary information reports NMR characterization of only two unsymmetrical decacyclene derivatives. Two others are reported with only HRMS characterization, and no others are reported.

We agree that the NMR characterisation is highly desired. However, the complexity and low solubility of final molecules make it difficult. For the partially soluble systems NMR is frequently uninformative because of the presence of stereoisomers (e.g 3 chiral centers in AAA, and six chiral centers in CCC), rotamers (biphenyl in the block A) and the aggregation phenomena. In this case the MS characterisation appears to be more informative. Therefore, we have carefully designed our blocks to avoid unambiguous interpretations of the MS data. As it mentioned in the manuscript - "*all segments were specifically designed to have a different number of carbon atoms (C₂₄H₁₄, C₂₆H₁₄ and C₂₈H₁₄ for A, B and C accordingly) allowing simple identification/characterisation of final precursors by MS analysis*". Therefore, a well established and reliably verified transformation was selected to perform the last synthetic step. Additionally, we provide unambiguous (X-ray single crystal analysis) structural confirmation of building blocks used in the assembly. According to MS data we provide direct evidence of the formation of 21 precursors, 12 of which were obtained in pure (individual) form as indicated by the HPLC/MS analysis (corrected to 10 since 2 were obtained as a mixture). All these data are presented in the

manuscript/supplementary information. Therefore, we would like to keep our claim: *“The possibility to derive required precursors in pure form was shown on 10 examples.”* Moreover, we were able to obtain acceptable ¹H-NMR spectra for four key precursors on which the multi-assembly concept is demonstrated (some of measurements were made on the sensitivity border).

3. The approach suffers from serious problems caused by product insolubility, which hampers (or precludes) isolation, purification, and characterization of the expected unsymmetrical decacyclene derivatives.

We agree that the low solubility of final product make the separation and characterisation extremely difficult. However, we were able to perform the separation of key precursors in multi milligram scale. All 12 compounds were obtained in pure form according HPLC/MS analysis.

4. Thus, the realization of this proposed concept is still in a very preliminary state and is hardly useable. No attempt to convert any of the synthesized "precursors" into the corresponding SWCNTs is reported.

The main goal of this work was to find a facile synthetic pathway to elusive SWCNT seeds. However, following Reviewer's suggestion, the demonstration of the SWCNT formation from one selected molecular seed was included in the revised version.

5. Schema 1 outlines a considerable amount of synthetic organic chemistry that the authors performed to access the substituted acenaphthenones for use in the acid catalyzed aldol cyclomerization reactions. Although such work did indeed provide the desired compounds, the chemistry used is not at all novel. The authors do not hold up this part of the work as a main selling point.

We would like to underline the power of our approach which lies in the fact that these simple blocks could provide access to very complex seeds for 25 different chiralities in just one synthetic step. As it correctly mentioned the chemistry used for the synthesis of blocks is not novel, and therefore is discussed only briefly.

6. Two points should be corrected when the authors revise this manuscript:
 - a) In the Results and discussion, line 2: the authors say "...in order to form strainless system." In reality, even IPR systems are not "strainless"

Accepted. The term „strainless system“ is changed to the “less strained system”

- b. In discussing the combinatorial trimerization of A and B, the authors say "...showing the formation of all desired mixed A/B trimers" They should say both of the desired mixed A/B trimers, since only two of the trimers are mixed.

Accepted. Mixing of 2 blocks provide four molecules which are precursor for specific chirality and all four are desired. However, we agree that only two of them can be called as mixed. This statement was corrected to “showing the formation of all desired trimers”

Reviewer #2 (Remarks to the Author):

Tomada et al reported the synthesis of SWCNT precursors by aldol-type trimerization. Although the synthetic route is not new, I think this work is an important paper that will have a large impact on the field of carbon-based materials. As a background, they accomplished controlled synthesis of single chirality CNT (Nature 2014). They synthesized C₉₆H₅₄ precursor, which was treated with surface-catalyzed cyclodehydrogenation. Then, epitaxial elongation of ultrashort nanotubes provided single chirality CNT. In order to extend this methodology, various precursors are naturally required. The overall synthesis is classical that utilizes methyl group on the aromatic ring as a handle for further functionalization to make the corresponding ketones, ie. radical benzylic bromination/cyanation/hydration/Friedel-Crafts reaction. The ketones are treated under acidic conditions for trimerization. Under some conditions, the tetramers were obtained. Note that such aldol-type trimerization that form benzene ring is well-known. Thus, this synthesis is essentially not new from the synthetic point of view. However, I think this work nicely established the route for various SWCNT precursors, which eventually opened up the possibility for programmed preparation of a series of single chirality SWCNT. Therefore, I recommend the manuscript to be published in Nature Comm. as a communication, which would attract not only organic chemist but also researchers who are working on carbon-based materials.

We thankful fort the highly positive evaluation of our work made by Reviewer #2

In the paper, there are many typos and mistakes. The authors should check carefully and correct them before submission.

The manuscript was carefully revised and respectively corrected

REVIEWERS' COMMENTS:

Reviewer #1 (Remarks to the Author):

This revised manuscript is much improved over the original manuscript, and the work has been extended in an important way through a collaboration with Professor Fasel. Overall, the contribution represents an important step forward in the quest for methods to synthesize uniform, chirality-pure, single-walled carbon nanotubes (SWCNTs), and publication in Nature Communications is warranted.

That said, the authors have unnecessarily exaggerated the extent of their contribution, both in the manuscript and in their rebuttal letter. In the concluding paragraph of the manuscript, the authors say, "...we demonstrate the synthesis of precursors for 21 different SWCNT chiralities using just three simple segments. The possibility to derive the required precursors in pure form was shown for 10 examples." Although technically true, many of the examples counted are homotrimers or homotetramers of a single ketone (e.g., AAA, BBB, AAAA, and BBBB), which were already accessible by known methods and do not demonstrate the importance of the authors' new "Combinatorial Design of Molecular Seeds." By my count, only six unsymmetrical precursors, which do demonstrate the importance of the authors' new combinatorial design of molecular seeds, were isolated in pure form (AAB, ABB, AAC, BBC, AAAB, and ABBB). The emphasis should be on these "mixed" trimers.

In the authors' rebuttal letter, the claim is made under point #1 that they "...have included the demonstration of the SWCNT growing from the one of synthesised precursor." The claim is repeated in point #4 with the claim of "...the demonstration of the SWCNT formation from one selected molecular seed was included in the revised version." In fact, the authors do not report the synthesis of any SWCNT from any of their molecular seeds. What they have added is new work of Professor Fasel, which demonstrates the successful conversion of one "mixed" trimer into a metal-bound end-cap of a SWCNT by surface-catalyzed cyclodehydrogenation (new Figure 6 and the accompanying text). This end-cap was NOT "grown" into a chirality-pure, SWCNT.

I recommend one addition to the manuscript that would increase its value, especially to the non-chemist readers. It would be very helpful if the authors would add a table listing in the first column the "precursors for 21 different SWCNT chiralities" that they have prepared (AAB, ABB, etc.), the chirality of the SWCNT for which each could serve as a seed for growth (i.e., the n,m index) in the second column, and whether or not the precursor was obtained in pure form in the third column. Are the chiralities of the SWCNTs that can formally be derived from these 21 precursors indeed all different, as claimed?

As a technical/linguistic point, the authors should use the terms "cyclotrimerization" and "cyclotetramerization" throughout the manuscript, instead of the more ambiguous terms "trimerization" and "tetramerization" they now use, which can be confused with the formation of acyclic trimers and tetramers.

Minor issues:

1) Page 1, lines 33/34: A citation to how "the so called chiral indices n and m" are defined should be included.

3) Page 3, lines 85/86: The authors say, "...the aldol trimerization of cyclic PAH ketones (figure 1) which was proven to be a prolific approach to extra-large PAH-precursors..." This is true, and numerous literature citations are provided. A citation to the most detailed discussion of this reaction should also be added: "The Synthesis of Tris-Annulated Benzenes by Aldol Trimerization of Cyclic Ketones," Chapter 1 in Modern Arene Chemistry, Astruc, D., Ed.; Wiley: New York, 2002.

4) Page 6, line 158: The word "motive" should be replaced by "motif"

- 5) Scheme 1 caption: "m) 2-Biphenyl acid bis(pinacol) ester" should be "2-biphenyl boronic acid bis(pinacol) ester"
- 6) Schema 1: The product shown for step s has the methyl group attached in the wrong place.
- 7) Page 11, lines 274/275: The authors propose using the abbreviation AABB to indicate the mixture of two isomers that have the same composition (AABB and ABAB). A more accurate and less misleading abbreviation would be A2B2.
- 8) Page 11, line 266: The citation to the literature given for the reaction conditions (reference 22) is incorrect and does not agree with the literature citation given in the Supporting Information.
- 9) In the Supp. Info., page 20, the authors are too brief in their describing the conditions for this key reaction as "The tetramers were obtained using previously described protocol." They should provide a detailed protocol here (in addition to the literature citation).
- 10) In the Supp. Info., terms such as DCM, DBOP, TBAB, Pd(dppf)Cl₂, which may be unfamiliar to non-chemists, should be spelled out at least once.
- 11) In the Supp. Info., the term "plugged through silica" is likely not clear to nonspecialists and should be stated more properly.
- 12) In the Supp. Info., mass spectral data should be provided as part of the characterization data for synthetic intermediates S11 and S15.
- 13) In the Supp. Info., in the synthesis of compound S12, for how long was the irradiation performed?
- 14) In the Supp. Info., page 17: The term "Trimerization" should be replaced by "Cyclotrimerization"
- 15) In the Supp. Info., page 19: HRMS data are reported for the two symmetrical cyclic tetramers, but only low resolution mass spectral data are provided for the "mixed" cyclic tetramers. HRMS data should be provided for all five cyclic tetramers.

Answer to Referees

Point-by-point response to Referees

(Reviewers' comments are highlighted in blue)

Reviewer #1:

1. This revised manuscript is much improved over the original manuscript, and the work has been extended in an important way through a collaboration with Professor Fasel. Overall, the contribution represents an important step forward in the quest for methods to synthesize uniform, chirality-pure, single-walled carbon nanotubes (SWCNTs), and publication in Nature Communications is warranted.

We thank the Reviewer #1 for positive evaluation of our work.

2. In the concluding paragraph of the manuscript, the authors say, "...we demonstrate the synthesis of precursors for 21 different SWCNT chiralities using just three simple segments. The possibility to derive the required precursors in pure form was shown for 10 examples." Although technically true, many of the examples counted are homotrimers or homotetramers of a single ketone (e.g., AAA, BBB, AAAA, and BBBB), which were already accessible by known methods and do not demonstrate the importance of the authors' new "Combinatorial Design of Molecular Seeds." By my count, only six unsymmetrical precursors, which do demonstrate the importance of the authors' new combinatorial design of molecular seeds, were isolated in pure form (AAB, ABB, AAC, BBC, AAAB, and ABBB). The emphasis should be on these "mixed" trimers.

We would like to keep this statement since all 21 precursors were generated and detected by the MS analysis. However, we agree that only 6 unsymmetrical precursors were isolated in pure form. This is additionally mentioned in the revised version.

3. In the authors' rebuttal letter, the claim is made under point #1 that they "...have included the demonstration of the SWCNT growing from the one of synthesised precursor." The claim is repeated in point #4 with the claim of "...the demonstration of the SWCNT formation from one selected molecular seed was included in the revised version." In fact, the authors do not report the synthesis of any SWCNT from any of their molecular seeds. What they have added is new work of Professor Fasel, which demonstrates the successful

conversion of one “mixed” trimer into a metal-bound end-cap of a SWCNT by surface-catalyzed cyclodehydrogenation (new Figure 6 and the accompanying text). This end-cap was NOT “grown” into a chirality-pure, SWCNT.

We agree that our statement in the rebuttal letter can be easily misunderstood. We have demonstrated the formation of ultra-sort SWCNT from one selected precursor. In the manuscript we use term CNT cap and/or SWCNT seed to avoid any misunderstanding.

4. I recommend one addition to the manuscript that would increase its value, especially to the non-chemist readers. It would be very helpful if the authors would add a table listing in the first column the “precursors for 21 different SWCNT chiralities” that they have prepared (AAB, ABB, etc.), the chirality of the SWCNT for which each could serve as a seed for growth (i.e., the n,m index) in the second column, and whether or not the precursor was obtained in pure form in the third column. Are the chiralities of the SWCNTs that can formally be derived from these 21 precursors indeed all different, as claimed?

The respective Table is added to Supplementary Materials

5. As a technical/linguistic point, the authors should use the terms “cyclotrimerization” and “cyclotetramerization” throughout the manuscript, instead of the more ambiguous terms “trimerization” and “tetramerization” they now use, which can be confused with the formation of acyclic trimers and tetramers.

Accepted and corrected.

Minor issues:

1) Page 1, lines 33/34: A citation to how “the so called chiral indices n and m” are defined should be included.

corrected

3) Page 3, lines 85/86: The authors say, “...the aldol trimerization of cyclic PAH ketones (figure 1) which was proven to be a prolific approach to extra-large PAH-precursors...” This is true, and numerous literature citations are provided. A citation to the most detailed discussion of this reaction should also be added: “The Synthesis of Tris-Annulated Benzenes by Aldol Trimerization of Cyclic Ketones,” Chapter 1 in Modern Arene Chemistry, Astruc, D., Ed.; Wiley: New York, 2002.

Reference is added

4) Page 6, line 158: The word “motive” should be replaced by “motif”

corrected

5) Scheme 1 caption: “(m) 2-Biphenyl acid bis(pinacol) ester” should be “2-biphenyl boronic acid bis(pinacol) ester”

corrected

6) Schema 1: The product shown for step s has the methyl group attached in the wrong place.

corrected

7) Page 11, lines 274/275: The authors propose using the abbreviation AABB to indicate the mixture of two isomers that have the same composition (AABB and ABAB). A more accurate and less misleading abbreviation would be A2B2.

Suggestion is accepted.

8) Page 11, line 266: The citation to the literature given for the reaction conditions (reference 22) is incorrect and does not agree with the literature citation given in the Supporting Information.

corrected

9) In the Supp. Info., page 20, the authors are too brief in their describing the conditions for this key reaction as “The tetramers were obtained using previously described protocol.” They should provide a detailed protocol here (in addition to the literature citation).

corrected

10) In the Supp. Info., terms such as DCM, DBOP, TBAB, Pd(dppf)Cl₂, which may be unfamiliar to non-chemists, should be spelled out at least once.

11) In the Supp. Info., the term “plugged through silica” is likely not clear to nonspecialists and should be stated more properly.

We believe that it is not necessary to explain these commonly used terms. Moreover these details are given exclusively for specialists in the field.

12) In the Supp. Info., mass spectral data should be provided as part of the characterization data for synthetic intermediates S11 and S15.

corrected

13) In the Supp. Info., in the synthesis of compound S12, for how long was the irradiation performed?

The reaction was monitored by TLC. This information is added in the revised version.

14) In the Supp. Info., page 17: The term “Trimerization” should be replaced by “Cyclotrimerization”

corrected

15) In the Supp. Info., page 19: HRMS data are reported for the two symmetrical cyclic tetramers, but only low resolution mass spectral data are provided for the “mixed” cyclic tetramers. HRMS data should be provided for all five cyclic tetramers.

HRMS data are added

We thank the Reviewer #1 for very careful reading of the work and providing valuable and useful remarks and corrections which remarkably improve the quality of the manuscript.